# Vegetable Peeling: A Case Study in Constrained Dexterous Manipulation

**Abstract:** Recent studies have made significant progress in addressing dexterous manipulation problems, particularly in in-hand object reorientation. However, there are few existing works that explore the potential utilization of developed dexterous manipulation controllers for downstream tasks. In this study, we focus on constrained dexterous manipulation for food peeling. Food peeling presents various constraints on the reorientation controller, such as the requirement for the hand to securely hold the object after reorientation for peeling. We propose a simple system for learning a reorientation controller that facilitates the subsequent peeling task. Videos are available at: https://sites.google.com/view/food-peeling.

**Keywords:** Dexterous manipulation, In-hand object reorientation, vegetable peeling

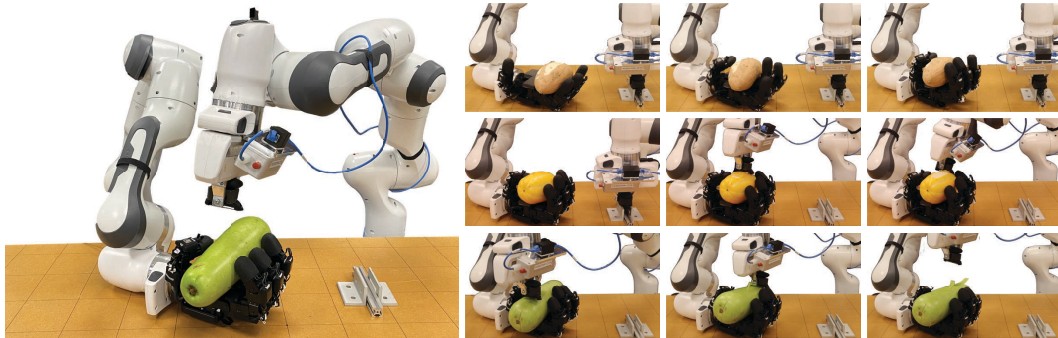

Figure 1: We present a dexterous manipulation system that utilizes an Allegro hand mounted on a Franka robot arm to reorient food items for downstream peeling. The other Franka robot arm uses its gripper to grasp a peeler for peeling. The reorientation controller for the Allegro hand is learned through reinforcement learning, while the peeling is performed via teleoperation. On the left of the figure, we show the whole system. On the right, from the top row to the bottom row, we illustrate the processes of reorienting a sweet potato, and peeling a melon and a squash.

## 1 INTRODUCTION

Having robots perform food preparation tasks has been of great interest in robotics. Imagine the scenario of making mashed potatoes, where a critical step is to peel potatoes. Humans peel potatoes by grasping the potato in one hand and using the second hand to actuate a peeler to remove the potato's skin. After a part of the potato is peeled, it is rotated while being held in the hand (i.e., *in-hand manipulation*) and peeled again. The sequence of rotating and peeling continues until all of the potato's skin is removed. In this work, we present a robotic system that can re-orient different vegetables using an Allegro hand in a way that their skin can be peeled using another manipulator. Our setup is shown in Figure 1.

In-hand rotation of vegetables is an instance of dexterous manipulation problem [1], a family of tasks that involves continuously controlling the force on an object while it is moving with respect to the fingertips [2, 3]. The challenges in dexterous manipulation stem from the frequent making and breaking of contact, issues in contact modeling, high-dimensional control space, perception challenges due to severe occlusions, etc. A body of work made simplifying assumptions such as manipulating convex objects [4, 5, 1, 6], small finger motions[7, 8, 9], slow or quasi-static motion or manipulating a few specific objects [10, 7, 8] to leverage trajectory optimization or planning-based methods to achieve in-hand object re-orientation [1, 7, 8, 9, 6, 4, 5, 10]. Another line of work has used reinforcement learning for in-hand re-orientation[11, 12, 13, 14, 15] and recent works have leveraged simulation to train policies capable of dynamically re-orienting a diverse set of new objects in real-time and in the real world [11, 12].

There are several challenges in adapting re-orientation controllers for a downstream task such as peeling vegetables. These challenges stem from the fact that controllers optimized for re-orientation [16, 13, 14, 15, 12] are only optimized to continuously reorient the object and not to satisfy numerous constraints arising from task-specific requirements. For instance, peeling vegetables requires the hand to **first** *stop* re-orienting the object and then for the peeler to peel the vegetable. Many prior works solve a version of the re-orientation problem where the object is continuously rotated [17, 16, 13] or otherwise perform quasistatic re-orientation [8]. Stopping and re-starting *dynamic* re-orientation is difficult due to the challenge of dealing with the object's inertia. **Second**, the hand needs to *hold* the object firmly enough to resist forces applied by the peeler. The closest work that attempts to hold the object at a target configuration [12] is only able to loosely hold the object which is insufficient for resisting forces. **Third**, the hand needs to reorient the vegetable *along a specific axis in place*. Here, the specific axis refers to the rotational axis on the object that is parallel to the peeling direction. Similar to how humans reorient vegetables for peeling, it is desirable for the hand to reorient the object in place so that multiple consecutive cycles of reorientation and peeling can be performed. If the object substantially shifts its position during reorientation, the controller will struggle to reorient and hold the object at future time steps. **Fourth**, when the vegetable is held stationary the fingers should *not obstruct* the top surface of the vegetable to ensure that the peeler can peel the vegetable.

While in-hand object reorientation has been widely studied [11, 12, 16, 18, 13, 17], no prior works can satisfy the constraints mentioned above. Yet, these constraints become critical for downstream dexterous manipulation beyond object re-orientation. We use vegetable peeling as a case study to investigate the challenges and solutions for building a dexterous manipulation system that can operate under constraints. We develop a framework where we leverage reinforcement learning in simulation to train a policy that can perform object re-orientation under constraints. For the peeling, we have a human teleoperate the peeler. Our contributions are as follows:

1. A framework for solving dexterous manipulation problems under constraints.

2. We propose a method that can make RL policy learn to stop its motion and hold objects firmly in hand – a critical behavior for many downstream dexterous manipulation problems.

3. We present a step towards a robotic system capable of peeling diverse vegetables with different shapes, masses, and material properties while holding and manipulating the vegetables in hand.

## 2 RELATED WORK

**In-hand Object Reorientation**: Dexterous manipulation involves the use of high degrees-of-freedom (DoF) manipulators for object manipulation [19]. Its requirement for high-dimensional real-time control and its nature of frequent contact-making and breaking present grand challenges to roboticists. Recently, there has been a growth of interest in a particular instance of dexterous manipulation problems: in-hand object reorientation. This problem is of particular interest as it is a necessary step in many tool-use scenarios. For example, to use a screwdriver for tightening a screw, one has to reorient the screwdriver to align it with the screw. We can cluster the works in in-hand object

reorientation from many aspects. For example, from the perspective of sensory information, [20] studies open-loop cube reorientation without using any sensors, [21, 5, 16, 10, 22] use motion capture system or special tracking markers for object reorientation, [17] uses proprioceptive sensors such as joint encoders, [23, 24, 15, 14] use tactile sensors and [25, 16, 12, 18] utilize vision sensors. In terms of the dynamics of the system, [7, 8, 9] achieved object reorientation under the assumption of quasi-static motion where object moves slowly and its inertia effect can be ignored, while [15, 16, 12, 14, 26] focuses on dynamic object reorientation where object is manipulated in a fast and dynamic way. To make in-hand object manipulation useful for downstream tool use tasks, one important aspect of the skill is the ability of stably and firmly holding the object in end of the policy rollout. While many prior works on dynamic manipulation such as [16, 10, 14, 15, 17] only consider endlessly rotating the object in hand and cannot stop the object stably when the object reaches the goal orientation, some works such as [12, 26] try to develop controllers that can reorient objects in hand and also hold the object in the goal orientation. Our work studies dynamic in-hand object manipulation with the capability of stopping objects stably in hand.

**Reinforcement Learning for Contact-rich Tasks**: Contact-rich tasks are particularly challenging due to the difficulty in modeling the system dynamics, especially when the tasks are performed in the wild, outside of a constrained and controlled setting. Examples to such tasks include quadruped robots hiking in mountains and robot hands reorienting various everyday objects. There have been many works using reinforcement learning to learn controllers for solving contact-rich tasks [27, 16, 13, 28, 29, 30]. In the real world, robots typically only have access to a limited amount of state information of the system due to the lack of sensors or the challenges in setting up the sensors. Using reinforcement learning to learn controllers from scratch with limited sensory information tends to be data-inefficient. One way to speed up policy learning is to provide asymmetric information to the policy and value function, where the value function observes much more privileged information [16, 13, 27, 31]. Another method is to decouple policy learning into two stages: a reinforcement learning stage where agents (teacher) observe privileged fully-observable state information, and an imitation learning stage where the policy with limited sensory observation input (student) learns to imitate the policy with fully-observable state information. This approach has been successfully applied to various contact-rich problems such as locomotion [32, 33, 30, 34, 35] and dexterous manipulation [11, 12, 17]. Our pipeline is built upon the idea of teacher-student policy learning and has made several key improvements, which we will detail below.

# 3 METHOD

Peeling requires a reorientation controller that can stop its motion and firmly hold objects after reorientation. The first step in stopping is to decide when re-orientation should be stopped. One possibility is to have a perception system predict the desired rotation angle after which the next round of peeling would be performed. To accomplish the goal, the robot would need to track changes in object pose and compare it with the target rotation angle. However, accurately estimating object pose is challenging, especially when generalization to new objects is necessary [36, 16, 13].

One of our insights is that instead of training a predictor for desired rotation angle and object pose estimation, it can be *easier* and *sufficient* to train a *binary vision classifier* that detects in real-time when the peeled part has been turned over. With such a classifier, the reorientation controller's job is simply to keep reorienting the object until it receives a stop signal. In this formulation, unlike prior works [11, 12], the reorientation controller is not conditioned on target orientation but rather on a stop signal. Formally, the policy takes as input a binary variable $I_t^{stop} \in \{0, 1\}$ representing the stop signal. If $I_t^{stop} = 1$, the policy should stop immediately and ensure the fingers stably and firmly hold the object. Otherwise, the policy should continue reorienting the object. Note that in this work, we focus on learning the reorientation controller, leaving integration of a vision classifier to future work.

The next question is how to train such a policy. Using RL to train the policy from scratch can be challenging and requires extensive reward shaping because $I_t^{stop} = 1$ is a rare event in an episode,

121 and when the $I_t^{stop}$ is flipped to one from zero, the policy needs to quickly stop the motion posing a
122 hard-exploration challenge.

123 Prior works [11, 12] show success in training a goal-conditioned object reorientation controller. Can
124 we leverage a goal-conditioned reorientation controller to train a controller that reacts to a stop signal?
125 It turns out we can formulate this using the teacher-student learning framework [11, 12, 37, 34, 33].
126 Specifically, we can use RL to train a goal-conditioned controller that reorients an object by random
127 goal angles along its rotational axis. This acts as the teacher. Next, we can use imitation learning
128 (specifically DAGGER [38]) to train a controller conditioned on the stop signal to imitate the teacher.
129 The stop signal can be generated during training by checking if the orientation distance to the goal is
130 below a threshold. Using imitation learning bypasses the hard exploration challenge.

## 3.1 Teacher Policy Learning: Reorient and Stop

132 We train the teacher policy to re-orient the object along a pre-defined axis and stop (see Figure 2a).
133 The teacher is formulated as a goal-conditioned policy $a_t^{\mathcal{E}} = \pi^{\mathcal{E}}(o_t^{\mathcal{E}}, a_{t-1}, g)$, where $\mathcal{E}$ represents
134 variables for the teacher policy, $o_t$ is the observation, $a_t$ is the action command, $g$ is the goal
135 representing the amount by which the object needs to be re-oriented. $g$ is randomly and uniformly
136 sampled from $[1.57, 4.0]$rad during training.

137 While the teacher policy's formulation is similar to that in prior works [11, 12], we propose (i) a
138 much simpler reward function, (ii) new success criteria that effectively encourages the policy to stop
139 the object and firmly hold it, and (iii) an interpolation scheme that enables smoother policy actions in
140 the real world.

### 3.1.1 Observation and Action Space

142 $o_t^{\mathcal{E}}$ includes joint positions and velocities, the fingertip poses and velocities, object pose and velocity,
143 the distance between the current object orientation and the goal orientation, and whether any of the
144 fingertips touch the object. $a_t$ is the delta joint position command. The neural network policy runs at
145 12Hz.

### 3.1.2 Reward Function

147 For the task of in-hand re-orientation, we found a simple way to specify the reward function.
148 Specifically, we manually move the real Allegro hand to a good pose where the constraints mentioned
149 above are satisfied (e.g., the fingers do not cover the food item), and the fingers touch the object and
150 are ready to reorient it. We record the joint positions as $q^{demo}$. During training in simulation, we
151 encourage the joint positions at any time step to be close to $q^{demo}$.

152 Overall, our reward function is as follows:

$$r_t = c_1 \mathbb{1}(\text{Task successful}) + c_2 \frac{1}{|\Delta\theta_t| + \epsilon_\theta} \tag{1}$$

$$+ c_3 \left\| q_t - q^{demo} \right\|_2^2 \tag{2}$$

153 where $c_1, c_2, c_3$ are coefficients. $\Delta\theta_t$ is the distance between the object's current and goal orientation.
154 The first two terms are task rewards for object reorientation. The last term is to regulate hand behavior.

### 3.1.3 Success Criteria

156 In a goal-conditioned object reorientation, a common way to claim the task successful is by checking
157 if the distance between the object's current and the goal orientation is smaller than a threshold value
158 (orientation criterion $C_{ori} = \Delta\theta < \bar{\theta}$) [16, 13]. Another criterion is that all the fingertips should
159 make contact with the object (contact criterion $C_{contact}$), a pre-requisite for firmly holding the object
160 after reorientation. However, only checking these two criteria is insufficient to ensure the policy
161 learns to stop the motion and hold the object firmly around the goal orientation, as discussed in [12].
162 The policy can oscillate around the goal state due to observation and control delay and noise.

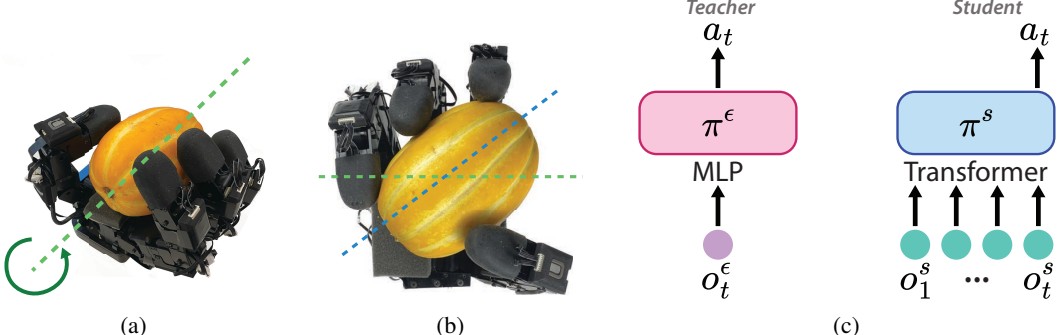

Figure 2: **(a)** shows an example of the rotational axis of a melon. **(b)** shows an example where the object's orientation (the blue line) has a large deviation from the desired rotational axis (the green line). We reset the episode when this occurs. **(c)** shows the policy Architecture for the teacher and the student. In this figure, we use $o_t$ to represent all the policy input at each time step.

To further encourage the policy to stop robot motion when the goal is reached and firmly hold the object, we propose adding time constraints to the success criteria: both $C_{ori}$ and $C_{contact}$ should be continuously satisfied for $\bar{T}^{succ}$ time steps. Adding this criterion makes the MDP partially observable since the policy's observation lacks the knowledge of time. Therefore, to facilitate policy learning, we augment the observation space with a scalar indicator variable $I^{succ} = t^{succ}/\bar{T}^{succ} \in [0, 1]$, where $t^{succ}$ is the number of consecutive steps satisfying $C_{ori}$ and $C_{contact}$. The observation space becomes $o^{\mathcal{E}} := o^{\mathcal{E}} \oplus I^{succ}$. In this work, $\bar{\theta} = 0.2$rad, $\bar{T}^{succ} = 8$.

## 3.2 Student Policy Learning: Imitate and Stop

After learning a goal-conditional teacher policy $a_t^{\mathcal{E}} = \pi^{\mathcal{E}}(o_t^{\mathcal{E}}, a_{t-1}, g)$, the next question is how to train a real-world deployable student policy that can rotate the object in hand and hold it stably after reorientation. We propose conditioning the student policy on a stop signal $I_t^{stop} \in \{0, 1\}$: $a_t^{\mathcal{S}} = \pi^{\mathcal{S}}(o_t^{\mathcal{S}}, a_{t-1}, I_t^{stop})$. In other words, the student policy should continue reorienting the object when $I_t^{stop} = 0$, but stably hold the object when $I_t^{stop} = 1$. This design choice provides flexibility in how we control the policy to stop the reorientation. For example, the policy could rotate the object for a pre-specified amount of time (i.e., set $I_t^{stop} = 1$ after $t$ seconds). Alternatively, an external perception module could detect when the peeled part has fully turned over, triggering $I_t^{stop} = 1$ and the policy to stop the motion and hold the object immediately.

How can we use the learned goal-conditioned teacher policy to train a student policy that is conditioned on the stop signal? We can set the value for $I_t^{stop}$ automatically during policy rollout based on the orientation distance $\Delta\theta_t$.

$$I_t^{stop} = \begin{cases} 0 & \text{if } \Delta\theta_t > \bar{\theta} \\ 1 & \text{otherwise} \end{cases}$$

### 3.2.1 Observation Space

In this work, we only use proprioceptive sensory information (joint positions $q_t$ and velocities $\dot{q}_t$) as the observation input ($o_t^{\mathcal{S}}$).

### 3.2.2 Policy Architecture

As the student policy only has access to a limited amount of sensory information (a POMDP setting), it is important to incorporate history information, as has been done in previous works [16, 13, 12]. While [16, 13, 12] utilized RNNs to process history information, Transformers [39] have gained significant attention due to their improved performance and faster training in domains such as natural language processing. Therefore, in this work, we employ a Transformer-based policy architecture. $a_t^{\mathcal{S}} = \pi^{\mathcal{S}}(o_1^{\mathcal{S}}, a_0, I_1^{stop}, ..., o_t^{\mathcal{S}}, a_{t-1}, I_t^{stop})$. The policy is a decoder-only attention

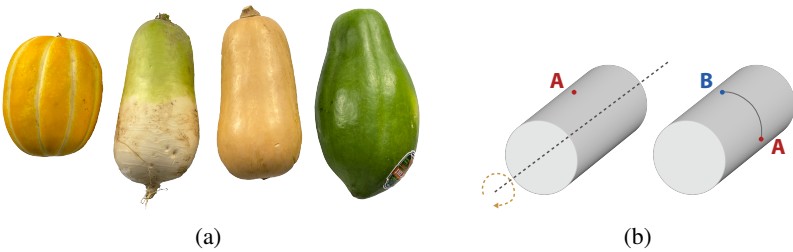

(a)                           (b)

Figure 3: **(a)** shows the objects for evaluation: melon, radish, pumpkin, papaya. **(b)** shows the traveling distance. Before reorientation begins, we ensure a reference point (point A) is facing upward. After reorientation, we identify the point (point B) now facing upward. We then measure the distance from point A to point B along the contour.

network (Figure 2c) with three self-attention layers. The hidden size is 256, the intermediate size is 512, and the number of attention heads is 8.

### 3.2.3  Training

The policy is trained using DAGGER [38].

### 3.3  Peeling via Teleoperation

We demonstrate that our reorientation controller can be used for downstream peeling tasks by teleoperating a Franka Panda robot arm to do the peeling. A 200 Hz operational space impedance controller [40] runs on the Panda arm, controlling for pose via torque, and an operator interacts with a Haption Virtuose™ 6D HF TAO[1] device. Bilateral position-position haptic coupling is done between the two devices. The controllers and haptic coupling are implemented using Drake [41].

## 4  RESULTS

To quantitatively evaluate the real-world policy transfer performance, we tested the controller on four vegetables (Figure 3a): a pumpkin (mass: 827g), a melon(623g), a radish(727g), a papaya(848g).

### 4.1  Traveling distance for a fixed amount of commanded motion time

The first question we want to answer is whether the learned policy can successfully reorient vegetables in the real world. In peeling, the width of the peeled part depends on the peeler's width. Thus, it is more informative to measure how much the reorientation controller rotates an object by the traveling distance of a surface point, rather than the absolute rotation angle. Specifically, we mark a reference point $P^{ref}$ on the object surface near the mid-point of its rotational axis. At the start, we ensure $P^{ref}$ is centered and facing upward when held. After reorientation, we record the new point $P^{new}$ that is now centered and facing upward. We then measure the contour length from $P^{new}$ to $P^{ref}$ along the surface (Figure 3b).

To demonstrate the capability of our controller to reorient real objects, we conducted two rounds of testing. Our controller is trained to stop motion when it receives a stop signal. In the first round, we sent the stop signal 3.5 seconds after the controller started rotating. In the second round, we sent the stop signal 7 seconds after start. We repeated each test 10 times. As shown in Figure 4a, the controller successfully reoriented all four food items by a sufficient amount for peeling. When commanded to reorient for 3.5s, 90% of tests reoriented the objects by at least 4cm. With 7s, 90% of tests reoriented objects by at least 7.3cm. Given more time, the controller reoriented objects by a larger amount.

---

[1]https://www.haption.com/en/products-en/virtuose-6d-tao-en.html#fa-download-downloads

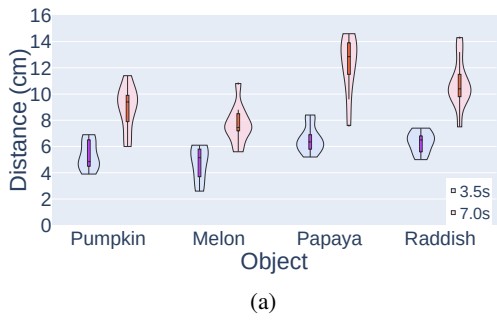 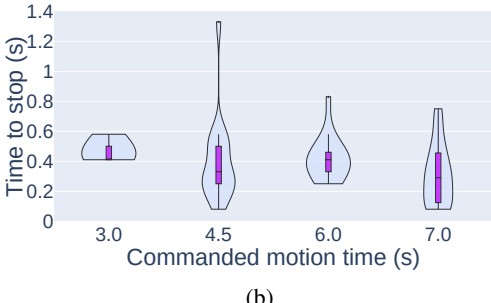

|       |       |
|-------|-------|
| (a)   | (b)   |

Figure 4: **(a)**: Violin plots showing the distribution of the traveling distance of a point on the object surface after the controller is commanded to rotate the object for 3.5 s and 7 s, respectively. **(b)**: Violin plot showing the distribution of time taken by the controller to transition from rotating the object in hand to firmly holding the object after receiving the stop signal. The $x$-axis represents the timing of the stop signal sent to the controller after it starts.

Table 1: Successful lifting rate (10 tests each)

| Commanded motion time | Pumpkin | Melon | Papaya | Radish |
|:---------------------:|:-------:|:-----:|:------:|:------:|
| 3.5s                  | 80%     | 90%   | 80%    | 90%    |
| 7s                    | 100%    | 90%   | 100%   | 90%    |

### 4.2 How well does the controller track the commanded motion time?

As discussed in Section 3, if our controller can quickly respond to a stop signal at any time step, it can be combined with a perception system that tracks peeling progress. Hence, we measured how long it takes to stop the hand and object motion after receiving the stop signal. As shown in Figure 4b, the motion stops after 0.4s on average after the controller receives the stop signal.

### 4.3 Firm grasp after reorientation

To enable downstream peeling, the reorientation controller must learn to firmly grasp the object after stopping finger motion. We tested this by checking if the Allegro hand and object could be lifted in the air for 3s by only lifting the object. Table 1 shows that across objects and commanded times, the controller firmly grasped objects in 90% of tests.

### 4.4 Real-world Peeling

We evaluated whether the reorientation controller could reorient food items to facilitate peeling (Figure 1). Testing showed that peeling applied substantial pulling forces on objects. However, in most cases, the hand maintained a firm enough grasp to enable successful peeling.

### 4.5 Ablation study

#### 4.5.1 Demo term in Reward function

We proposed using a keyframe demonstration to ease reward shaping. To evaluate its effectiveness, we compared learning curves of the teacher policies trained with and without the $c_3 \left\| \boldsymbol{q}_t - \boldsymbol{q}^{demo} \right\|_2^2$ reward term. As shown in Figure 5a, adding the keyframe substantially improved learning. Additionally, it demonstrates that mimicking the keyframe pose via a single reward term effectively reduces the reward-shaping burden.

#### 4.5.2 Transformer vs RNN

Different from prior works [16, 13, 11, 12], our student policy uses a Transformer architecture instead of an RNN architecture. We compared the learning performance of a Transformer-based policy and

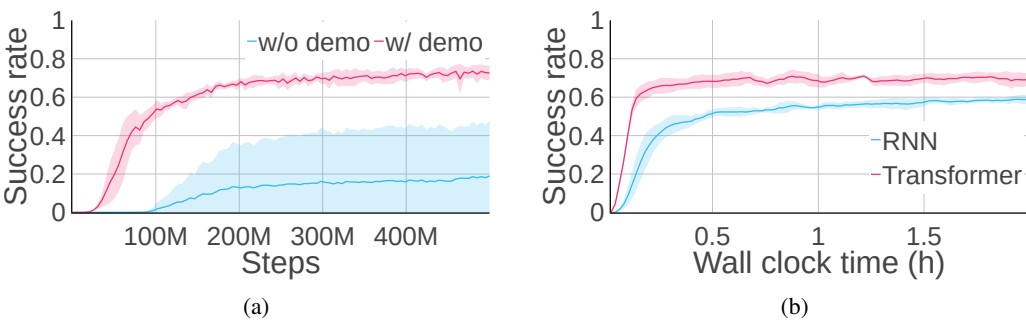

Figure 5: **(a)** shows learning curves of the teacher policies with or without $c_3 \left\| \boldsymbol{q}_t - \boldsymbol{q}^{demo} \right\|_2^2$ in the reward function. **(b)** shows the learning curve of student policies with a Transformer or RNN archtecture with respect to the number of samples.

an RNN-based policy. Figure 5b shows that a Transformer-based policy learns much faster and gets better performance at convergence than an RNN-based policy.

## 5 DISCUSSIONS

The reorientation controller described in this study is a blind controller that relies solely on proprioceptive sensory information. Although it has shown the ability to successfully reorient heavy objects and securely hold them in place, performance could potentially be enhanced by incorporating visual and tactile feedback. For instance, visual information could help prevent objects from falling. Additionally, future work could involve learning a peeling policy via behavior cloning on data collected via teleoperation to achieve full autonomy of the system.

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
