# OpenReview forum: "Vegetable Peeling: A Case Study in Constrained Dexterous Manipulation"
_robot-learning.org/CoRL/2023/Workshop/TGR — CoRL 2023 Workshop TGR Poster_

### Official Review · Reviewer_nLZf · 2023-10-16

**Rating:** 7
**Confidence:** 3

**Review:**

This paper presents a case study for constrained dexterous manipulation. Although this is probably a specific task, some methods proposed in this paper might be useful for other similar tasks.

---

### Decision · Program_Chairs · 2023-10-20

**Decision:**

Accept (Poster)

**Comment:**

Great paper!  Learning dexterous peeling skill is a important step towards generalist robot learning.